# Role of NADPH Oxidases in Stroke Recovery

**DOI:** 10.3390/antiox13091065

**Published:** 2024-08-30

**Authors:** Dong-Hee Choi, In-Ae Choi, Jongmin Lee

**Affiliations:** 1Center for Neuroscience Research, Institute of Biomedical Science and Technology, Konkuk University, Seoul 05029, Republic of Korea; inaes2@bu.ac.kr; 2Department of Medical Science, Konkuk University School of Medicine, Konkuk University, Seoul 05029, Republic of Korea; 3Department of Occupational Therapy, Division of Health, Baekseok University, Cheonan-si 31065, Republic of Korea; 4Department of Rehabilitation Medicine, Konkuk University School of Medicine, Konkuk University, Seoul 05029, Republic of Korea

**Keywords:** stroke, recovery, repair, oxidative stress, NADPH oxidases, angiogenesis, neurogenesis, functional recovery, stroke interventions, stroke with comorbidity

## Abstract

Stroke is one of the most significant causes of death and long-term disability globally. Overproduction of reactive oxygen species by NADPH oxidase (NOX) plays an important role in exacerbating oxidative stress and causing neuronal damage after a stroke. There is growing evidence that NOX inhibition prevents ischemic injury and that the role of NOX in brain damage or recovery depends on specific post-stroke phases. In addition to studies on post-stroke neuroprotection by NOX inhibition, recent reports have also demonstrated the role of NOX in stroke recovery, a critical process for brain adaptation and functional reorganization after a stroke. Therefore, in this review, we investigated the role of NOX in stroke recovery with the aim of integrating preclinical findings into potential therapeutic strategies to improve stroke recovery.

## 1. Introduction

Stroke ranks among the top causes of mortality and disability worldwide, significantly impacting socioeconomic costs. Ischemic stroke, which occurs due to the disruption of cerebral blood flow, leading to insufficient oxygen supply to affected areas, accounts for nearly 80–85% of all stroke cases [1,2]. The pathophysiology of stroke is complex and multifaceted, involving excitotoxicity, inflammation, and oxidative stress, all of which contribute to neuronal death and long-term disability [3,4,5].

Oxidative stress significantly influences the pathophysiology of brain damage leading to stroke, especially through the mediation of severe toxicity during the acute phase, resulting in apoptosis, inflammation, and senescence [6,7]. The state of oxidative stress occurs when the physiological antioxidant defenses are unable to counteract the production and accumulation of reactive oxygen species (ROS) [7,8]. Although antioxidants have shown promise in preclinical trials, translating these benefits to clinical settings has proven challenging [9]. Consequently, there is a growing interest in the therapeutic strategies that target oxidative stress, especially through the inhibition of NADPH oxidases (NOXs), enzymes primarily responsible for ROS production [10,11].

Recent research has highlighted the potential of NOX inhibitors in effectively reducing ROS production and preventing the brain tissue injury associated with ischemic stroke thus highlighting the therapeutic value of targeting NOXs for mitigating stroke-induced damage [10,11]. The efficacy of apocynin (APO), a NOX2 inhibitor widely used in experimental studies and various stroke models, is well established, and it has been shown to disrupt the translocation of the cytosolic component p47phox to the membrane subunit of the NOX complex [12]. Similarly, GKT-136901, which inhibits the activation of NOX1 and NOX4 [13], and ebselen, an inhibitor of both NOX1 and NOX2, have been shown to block the translocation of p47phox and p67phox, prevent the assembly of these NOX subunits, and offer neuroprotective benefits [12,14]. NOX2ds-tat is, another specific NOX2 inhibitor that, blocks p47phox binding and has demonstrated significant neuroprotective effects in stroke models [15,16].

Moreover, the recent development of small-molecule NADPH inhibitors has motivated numerous studies investigating their selectivity for specific subtypes, as well as their mechanism of action and specificity [11,15]. These studies may improve our understanding of how the selective inhibition of NOX subtypes can potentially improve treatment and recovery in stroke management by minimizing off-target effects and maximizing efficacy. However, research on the inhibition of brain damage by regulating NOX activity has primarily focused on neuroprotection following stroke, with relatively fewer studies on recovery after a stroke.

The purpose of this review is to provide insights from preclinical studies by examining the effects of NOX inhibition on post-stroke repair and functional recovery at different stages of recovery.

The stroke recovery and rehabilitation roundtable has proposed a recovery timeline for clinical studies and preclinical studies in stroke animal models, dividing the recovery process into the hyperacute (first 24 h), acute (1 to 7 days), early subacute (7 days to 3 months), late subacute (3 to 6 months), and chronic phases (after 6 months) [17].

In the hyperacute phase, biological processes such as apoptosis, inflammation, and scar formation occur following excitotoxicity, and endogenous plasticity mechanisms initiated during the acute phase extend into the subacute phase, leading to functional recovery [17]. However, the recovery timeline differs between humans and animal models, with the maximum recovery period typically lasting up to 3 months in humans, compared to around 4 weeks in animal models [18]. In some studies, the animal model of cerebral infarction defines the acute phase as the period up to 3 days post-stroke, the subacute phase as up to 7 days, and the chronic phase as the period thereafter [19,20].

Given these differences, our study focused on the period from 24 h to 4 weeks post-stroke in animal models, encompassing both the critical window for initiating recovery processes and the progression into the later stages of recovery.

To search for animal experimental studies, a comprehensive search of MEDLINE and PubMed databases was performed without restrictions on publication date or language, following the recommendations of the Systematic Review Center for Experimental Animal Research (SYRCLE) [21]. All articles were solicited from the databases using the searching terms, “rat or mouse”, “stroke recovery”, “stroke repair”, and “NADPH oxidase”. A keyword search for “rat/mouse/stroke/recovery/repair/NADPH oxidase” was conducted, utilizing both medical subject headings (MeSH) terms and all-fields terms. The full search details for all databases can be found in the Appendix A. Of the 21 articles identified in MEDLINE and PubMed (as of May 2024), 7 articles that did not meet the study criteria were excluded (2 in vitro studies and 5 articles on other diseases). Finally, 14 articles were included in this review. The publication dates of these articles ranged from 1 December 2004 to 14 May 2024.

## 2. NADPH Oxidases

### 2.1. NOX Structure, Isoforms, and Functions

NOXs are membrane-bound enzyme complexes primarily responsible for the production of ROS through the transfer of electrons from NADPH to oxygen [22,23]. The basic structure of a NOX enzyme includes a catalytic subunit that spans the cell membrane, with the N-terminal domain facing the cytoplasm and the C-terminal domain exposed to the extracellular or luminal space [22,23]. This structure allows NOXs to transport electrons across the membrane, reducing oxygen to superoxide or other reactive species [23,24,25]. The NOX family of enzymes is essential for a variety of biological processes, with each isoform playing specific roles depending on its tissue distribution and the type of reactive oxygen species it produces [22,26,27]. NOX1, NOX2, NOX3, and NOX5 are primarily involved in generating superoxide, which can be converted to other ROS, while NOX4, DUOX1, and DUOX2 mainly produce hydrogen peroxide [22,23].

These ROS play critical roles in cell signaling [23,28], host defense [22,25,29], regulation of gene expression [28,30], and redox biology [31,32], with implications in both normal physiology and pathophysiology, including roles in cardiovascular diseases [33,34], immune responses [35,36], and even brain repair mechanisms post-stroke [37,38]. Figure 1 and Table 1 provide a comprehensive overview of the structure, isoforms, and functions of NOX.

### 2.2. Activation and Regulation of NOX Isoforms

Table 1 presents the specific activators, organizers, and regulators involved in the activation and regulatory processes of each NOX isoform.

NOX1 activation begins when NOXA1 binds to the NOX1 complex, facilitated by the organizer protein NOXO1 [39,40]. This binding, along with the involvement of Rac1 GTPase and protein kinase C (PKC), results in the structural changes necessary for the enzyme to begin producing superoxide [41,42,43]. Similarly, NOX2 is activated when p67phox binds to the NOX2 complex, organized by p47phox. Rac2 GTPase and PKC further regulate the complex, with p40phox playing a role in stabilizing the active complex, ultimately leading to superoxide production, which is vital for immune functions [23,25,44]. NOX3 follows a similar activation pattern to NOX1, with NOXA1 and NOXO1 facilitating its activation and Rac1 GTPase ensuring proper function for superoxide production [45,46]. NOX4 differs in that it is constitutively active, meaning it does not require an external activator; instead, its activity is regulated through stabilization by p22phox and expression control by transforming growth factor beta (TGF-β) [47,48,49,50]. This leads to the continuous production of hydrogen peroxide, a signaling molecule. NOX5 is activated by calcium ions (Ca^2+^), which bind directly to the enzyme, triggering its activity. Calmodulin further modulates this activation, enabling the enzyme to produce superoxide [51,52,53]. For DUOX1 and DUOX2, calcium also serves as the activator, with DUOXA1 and DUOXA2 supporting their activation. p22phox stabilizes the complexes, leading to the production of hydrogen peroxide, crucial for antimicrobial defense [23,54,55].

Overall, the activation of NOX isoforms involves a precise and coordinated process where specific activators and organizers interact with the NOX complexes, inducing conformational changes that trigger the enzyme’s activity, highlighting their specialized roles in various physiological processes.

Table 2 provides an overview of the 14 selected NOX-related stroke recovery studies reviewed in this paper. The table summarizes key details such as the animal species, models, ages or weights, timepoints, interventions, and outcomes.

## 3. Mechanisms of NOX in Stroke Recovery

### 3.1. NOX and Angiogenesis in Post Stroke Repair

Emerging evidence highlights the pivotal roles of vascular signaling and angiogenesis in the role of NOX in brain repair. Findings from two critical studies have revealed that the temporal and spatial dynamics of post-stroke NOX2 and NOX4 expression are associated with vascular remodeling and recovery.

Taylor et al. investigated the temporal and spatial expression and activity of NOX subunits in the brain, along with the angiogenesis profiles, following transient cerebral ischemia induced by endothelin-1 in Wistar rats aged 10–12 weeks. Their findings demonstrated that vascular endothelial growth factor (VEGF) mRNA levels rose in the ipsilateral cortex and striatum starting from 6 h and continuing up to 28 days after the stroke occurred. Significantly, the expression of NOX2 mRNA was also increased substantially through the initial 7 days, whereas NOX4 mRNA levels increased from day 7 to day 28 post-stroke. Angiogenic blood vessels were found to co-localize with NOX2, suggesting that NOX2-generated superoxide could be crucial in initiating vessel formation during the early phase of vascular remodeling [56].

The results of a study by McCann et al., [57] using 8–12-week-old C57BL/6 mice in the ischemia-reperfusion-induced middle cerebral artery occlusion (MCAO) model, provide further evidence for the beneficial aspects of NOX2 in stroke recovery. They demonstrated that the genetic deletion of *Nox2* delayed neuronal loss progression at 24 and 72 h post-stroke. Specifically, the *Nox2* knockout animals exhibited a reduced density of the blood vessels in the ipsilateral core than in the contralateral control at 24 h post-stroke [57]. Collectively, these results highlight the importance of NOX2 in enhancing the revascularization of the injured brain, thereby promoting endogenous brain repair processes that depend on revascularization. However, since the measurement of recovery outcomes in this study was limited to the acute phase, verification of the role of NOX2 through additional long-term research may be necessary.

The role of NOX2 and NOX4 in promoting angiogenesis and neovascularization after stroke involves the activation of NOX2 and NOX4, which increases the production of ROS. These ROS act as signal molecules, activating hypoxia-inducible factor 1 (HIF-1), which in turn upregulates the expression of vascular endothelial growth factor (VEGF). VEGF then binds to its receptor, VEGF receptor2 (VEGFR2), on the endothelial cells, initiating the process of angiogenesis or the formation of new blood vessels [70,71]. This angiogenesis ultimately leads to neovascularization, restoring blood flow and supporting the recovery of the brain after a stroke (Figure 2).

### 3.2. NOX and Neurogenesis in Post Stroke Repair

Neurogenesis can be an important mechanism for recovery following ischemic stroke. Various treatments for ischemic stroke have been shown to promote post-stroke brain repair and functional recovery in conjunction with neurogenesis, highlighting its therapeutic potential [72]. Endogenous neurogenesis contributes to brain repair, but it is not sufficient because of the high mortality of new neurons [73]. Therefore, it is necessary to identify a microenvironment that promotes the survival of these new cells during ischemic neurogenesis. Choi et al. explored the impact of NOX1 on the survival of these new neurons following a stroke [58].

Using the MCAO model in 8-week-old Wistar rats, they observed a significant increase in NOX1 expression as well as increased levels of oxidative markers such as 8-oxo-2′-deoxyguanosine in the cortical neurons surrounding the infarct lesion. The recovery outcomes, measured at 2, 7, 14, and 28 days after MCAO-reperfusion, showed that the use of RNA interference to reduce NOX1 expression not only significantly reduced infarct size and neuronal death but also significantly improved motor function recovery. This suggests a pivotal role for NOX1 in the regulation of oxidative damage [58].

Additionally, NOX1 inhibition increased both the survival and maturation of neural progenitor cells in the peri-infarct area, demonstrating the important role of NOX1 in regulating ischemic neurogenesis and subsequent recovery. Specifically, this study showed that *Nox1* knockdown via RNA interference significantly increased the population of cells expressing the cell proliferation marker bromodeoxyuridine (BrdU) and the progenitor cell marker doublecortin within the ipsilateral subventricular zone. Additionally, the increase in cells co-labeled with BrdU and the neuronal nuclei (NeuN) protein, a marker of mature neurons, was accompanied by a decrease in the aggregation of apoptotic newborn cells, as indicated by BrdU and active caspase-3 labeling. These results suggest that NOX1 inhibition after ischemia may positively affect the localization and proliferation of progenitor neurons after stroke and enhance the viability of neurons differentiated from these progenitor cells [58].

Figure 3 illustrates the role of NOX1 in neurogenesis following a stroke. The survival rate of neural progenitor cells (NPCs) is significantly reduced by the NOX-induced ROS after an ischemic stroke. However, the inhibition of NOX1 enhances the proliferation, migration, and survival of NPCs, leading to an increase in newly born mature neurons. This improvement in the neurogenesis mechanism ultimately promotes functional recovery.

### 3.3. NOXs in Post-Stroke Functional Recovery

Several studies have investigated the effects of modulating NOX activity on functional recovery after stroke. Using various experimental models, including mouse and rat MCAO- and ET-1-induced ischemic models, as well as neonatal hypoxia, these studies employed techniques such as genetic knockout/knockdown and APO treatment to inhibit NOX activity, with APO treatment administered generally within the range of 1 h before MCAO to 48 h post-reperfusion. The outcomes were measured between 24 h and 4 weeks post-stroke, focusing on the roles of NOX1, NOX2, and NOX4 in the recovery process.

Choi et al. [58] examined the effects of *Nox1* knockdown using an MCAO model in 4-week-old Wistar rats. When NOX1 expression was reduced via adeno-associated virus-mediated knockdown, there was a significant enhancement in the motor function recovery compared to the MCAO control group. Recovery outcomes were evaluated over 4 weeks using parallel bar walking tests, and the reduction in NOX1 expression was shown to positively influence post-stroke functional recovery.

Similarly, Qin et al. [59] demonstrated that the combined treatment of NADPH and APO, administered 8 h after reperfusion, had positive effects on post-stroke recovery in an MCAO model using ICR mice (25–30 g). They found that this combined treatment significantly reduced ischemia/reperfusion-induced brain inflammation and neuronal damage. NADPH plays a dual role, acting as a coenzyme for glutathione production or as a substrate for NOX, which generates ROS. In their study, the combination of NADPH and APO significantly attenuated the increase in NOX2, NOX4, and ROS levels, inhibited NF-κB signaling pathway activation, and suppressed the expression of inflammasome proteins, ultimately reducing infarct volume. Recovery outcomes were measured at 2, 7, 14, and 28 days, with the combined treatment, resulting in improved motor function as indicated by the rotarod test and enhanced cognitive function as demonstrated by the Y-maze task. Additionally, this treatment led to an increase in the long-term survival rate, highlighting its potential as an effective intervention for stroke recovery and rehabilitation. These findings suggest that NADPH has anti-inflammatory properties beyond its antioxidant effects, and that its combination with NOX inhibitors can enhance neuroprotection and functional recovery in ischemic stroke. However, the administration of APO 8 h after reperfusion in the acute phase represents a limitation of this study.

However, a recent study from 2022 by Yingze et al. [60] highlighted the complex and time-dependent roles of NOX2 in stroke recovery. Using an MCAO model in C57BL/6 mice (25–30 g, 8–10 weeks), APO was administered at 8 h post-MCAO/reperfusion, and the recovery outcomes were measured at 3, 7, 10, and 14 days. The observed ROS and NOX2 expression levels after MCAO reached their peak on day 3, but this effect gradually decreased over time; moreover, the administration of APO significantly suppressed these effects. The results showed that the APO treatment reduced infarct volume at 3, 5, and 14 days; however, it effectively decreased neurological deficit scores, motor impairment as assessed by the rotarod test, and mortality only at 3 days, while mortality and functional deficits increased between days 5 and 14. This study revealed that while ROS activity exacerbates brain damage during the acute phase following a stroke, it promotes recovery during the delayed phase through mechanisms involving neuroinflammation, autophagy, angiogenesis, and the phosphoinositide 3-kinase (PI3K)/Akt signaling pathway. On day 3 post-stroke, excessive ROS generation leads to the overproduction of autophagosomes, which in turn activates the NOD-like receptor family pyrin domain containing 3 (NLRP3) inflammasome, mediating the inflammatory responses and inhibiting the vascular growth. This process exacerbated the brain damage caused by the stroke, but the APO treatment effectively inhibited these harmful mechanisms. However, during the delayed phase between days 5 and 14, lower levels of NOX2-induced ROS may act as signaling molecules that stimulate moderate autophagosome production, thereby suppressing NLRP3 inflammasome activation and promoting angiogenesis through the activation of the PI3K/Akt/NF-kB signaling pathway.

This study demonstrated that while NOX inhibition has therapeutic effects during the acute phase, it excessively suppresses ROS, which serve as essential signaling molecules that are naturally reduced during the delayed phase, thereby interfering with critical recovery mechanisms. This disruption led to increased NLRP3 inflammasome activation, altered autophagy conditions, and suppressed angiogenesis, ultimately resulting in adverse outcomes.

The differences in recovery outcomes at the same time points between the studies by Qin et al. [59] and Yingze et al. [60] may be influenced by the use of different mouse strains—ICR and C57BL/6—despite identical stroke induction methods, reperfusion timing, and APO administration protocols. This study highlights the complex and time-dependent function of NOX2-mediated ROS in stroke recovery mechanisms and suggests the importance of a balanced approach to targeting NOX2 activity. To achieve precise NOX modulation, it is essential to first monitor the ROS levels using biomarkers, as the ROS levels vary at different stages post-stroke.

Additionally, studies have shown that the modulation of NOX activity can have different effects depending on age [61,62].

Age-related differences in the effects of NOX modulation were explored by comparing the effects of APO administered as a pre-treatment 30 min before MCAO, which involved the injection of an autologous blood clot directly into the MCA, followed by the administration of tPA (5 mg/kg) two hours later, in young adult (3–4 months old) and aged (18–20 months old) Sprague–Dawley rats [62]. Neurological function, blood–brain barrier (BBB) permeability, ischemic infarct volume, edema formation, and oxidative stress were assessed after inducing MCAO and tissue plasminogen activator-induced reperfusion [62]. In this model, APO reduced the infarct volume and had positive effects on recovery in the young rats, but in the aged rats, it increased infarct volume and resulted in negative recovery outcomes. Recovery outcomes were measured 24 h post-MCAO, with aged rats showing more severe brain damage and sustained functional deficits compared to their younger counterparts. Following MCAO, increased BBB permeability was also observed in the ipsilateral hemispheres of the young rats and was attenuated by APO treatment. However, the aged rats showed no change in BBB permeability after APO treatment and demonstrated increased BBB permeability in the contralateral hemisphere. The geriatric rats also had a marked reduction in antioxidant capacities, such as superoxide dismutase and glutathione peroxidase activities, compared to their younger counterparts. The outcomes from the aged animal model not only revealed exacerbated brain damage and sustained functional deficits following a stroke but also disclosed alterations in the contralateral hemisphere [62].

These findings emphasize the critical need for recognizing age-associated processes in stroke research and highlight the necessity of developing NOX modulation strategies specifically designed to accommodate these age-related dynamics. However, it is important to note that the outcomes in this study were measured only 24 h post-MCAO, which limits the understanding of the long-term effects of APO treatment on stroke recovery.

Fabian et al. [61] investigated the effects of APO following hypoxic–ischemic injury in a model using 7-day-old neonatal rats. APO was administered 2 h after carotid artery occlusion, and the recovery outcomes were measured at 24 and 72 h post-treatment. The study found that APO reduced the superoxide levels and restored vascular reactivity, suggesting that decreased NOX activity plays a crucial role in normalizing abnormal ROS concentrations during hyperoxic resuscitation after neonatal hypoxic–ischemic injury.

In detail, the loss of vascular reactivity induced by hypoxia–ischemia is partly due to the diminished production of nitric oxide, which is generated by nitric oxide synthase 3 (NOS3), an endothelial NOS and a critical determinant of vascular tone and function [61,74,75]. Additionally, superoxide production by NOXs in the endothelial cells is thought to play a crucial role in establishing vascular tone in response to various stimuli [24,27]. Superoxide can react with nitric oxide, reducing its concentration, or with tetrahydrobiopterin (BH4), the cofactor of NOS3, to alter its activity or uncouple it to, further increase superoxide production [61,76].

Furthermore, following neonatal hypoxic–ischemic injury, hyperoxic resuscitation reduces cerebral blood flow, decreases perivascular nitric oxide production, and increases superoxide levels. APO treatment can reduce elevated superoxide levels and nitric oxide production, as evidenced by the fluorescence measurements when using dihydroethidium (DHE) and 4,5-diaminofluorescein diacetate, respectively [61]. This suggests that reducing NOX activation normalizes aberrant radical production by altering ROS concentrations and inhibiting the uncoupling of endothelial NOS thus highlighting the important role of NOX in the regulation of vascular responses during hyperoxic resuscitation after neonatal hypoxic–ischemic injury.

Taken together, these studies highlight the importance of investigating the role of NOXs in stroke recovery across different age groups and developing more effective, age-appropriate interventions.

There is evidence suggesting that the impact of NOX on post-stroke recovery may vary depending on the cell type. Weston et al. [63] investigated the effects of the NOX2 inhibitor APO in an endothelin-1-induced MCAO model in conscious 10–12-week-old Wistar rats. APO (50 mg/kg) was administered 1 h before injury, with additional doses given at 24 and 48 h. By administering APO, the cortical infarct volume created after stroke was reduced by approximately 60%. However, this intervention neither inhibited striatal damage nor attenuated the neurological deficits measured 72 h after stroke, showing a selective protective effect against cell damage in specific brain regions. The study revealed that APO reduced ROS production by approximately 51% in microglia but increased ROS levels by approximately 27% in neurons. This suggests that NOX inhibition may have opposite effects depending on the cell type. These researchers suggested that these opposite effects of APO in neurons are probably due to the absence of MPO in neurons, highlighting the need for further studies [63]. These findings clearly highlight the importance of considering cell-specific differences when applying NOX inhibitors for stroke recovery and suggests the requirement of a more detailed approach that accounts for diverse responses across different cell types.

While this study demonstrated that APO pre-treatment and acute-phase administration could reduce cell death in specific peri-infarct regions and provided insights into the cell-type-specific mechanisms of NOX regulation, it did not ultimately improve neurological outcomes. Therefore, it appears that NOX2 inhibition may not provide a beneficial effect on functional recovery in the ET-1-induced MCAO rat model. Additionally, this study is limited by the fact that APO’s effects were measured only up to 72 h, highlighting the need for further long-term research. Considering that this study is the only one in this review that tested APO in the ET-1-induced MCAO rat model, future research should analyze the results of multiple studies conducted under the same model conditions.

## 4. Role of NOX in Stroke Interventions

Recent studies have highlighted the significance of NOX modulation in various stroke intervention strategies.

For instance, Shen et al. [64] investigated the impact of the timing of rehabilitation exercises on brain damage after ischemic stroke, with a particular focus on the role of NOX activation. In their study using adult male Sprague–Dawley rats subjected to 2 h of MCAO, it was found that the timing of exercise initiation significantly influenced NOX activation and its subsequent effects on the brain. The rats were divided into groups that began exercise either early (24 h post-reperfusion) or relatively late (3 days post-reperfusion), with a control group that did not perform any exercise. The study measured lactate production as an indicator of hyperglycolysis, NOX activity, and the expression of NOX subunits (p47phox, gp91phox, and p67phox), and glucose transporters at different time points. Post-stroke, increased lactate production, Glut-1 and Glut-3 expression, NOX activity, and NOX subunit expression were decreased in the groups that initiated exercise 24 h and 3 days after reperfusion [64].

This suggests that suppressing NOX expression through physical exercise may promote beneficial outcomes in stroke recovery. However, a significant limitation of this study is the lack of post-exercise recovery outcome measures, such as functional assessments or long-term recovery indicators. Although this study provides insight into cellular mechanisms of exercise therapy, such as apoptosis and NOX expression, it does not directly address whether exercise ultimately improves stroke recovery.

Another study by Jung et al. [65] explored the protective mechanisms of electroacupuncture preconditioning against ischemic stroke, focusing on its effects on BBB integrity and the reduction in oxidative stress through the modulation of NOX4 expression. Electroacupuncture, which combines traditional acupuncture with electrotherapy, has been increasingly used as a supplementary treatment for ischemic stroke due to its potential in mitigating BBB dysfunction and brain edema associated with ischemia-reperfusion injury [65,77]. In a transient MCAO model in adult C57BL/6 mice (20–25 g), the study found that electroacupuncture preconditioning, administered at specific acupoints (Baihui and Dazhui) for 20 min daily over 3 days prior to the ischemic event, led to enhanced neurological function and a significant reduction in infarct volume 24 h after ischemia [65]. The study also observed a marked reduction in Evans blue leakage and brain water content, indicating decreased BBB permeability and brain edema. Furthermore, electroacupuncture preconditioning effectively reduced the expression of astrocytic aquaporin 4, which is implicated in BBB disruption and the diminished ROS levels in brain tissues following ischemic injury. Notably, electroacupuncture selectively decreased NOX4 rather than NOX2 expression, suggesting a targeted mechanism by which it protects the BBB [65].

Despite these promising findings, the study’s reliance on preconditioning limits its clinical relevance, as this approach does not directly translate into post-stroke therapeutic interventions. Moreover, since all measurements were taken 24 h after ischemia, the study lacks long-term observations critical for fully understanding the impact of NOX suppression on stroke recovery. Further research is required to investigate the long-term effects of electroacupuncture and its potential application as a therapeutic intervention post-stroke.

Furthermore, the role of NOXs in stroke interventions also extends to pharmacological treatments. Park et al. explored the interaction between NOXs and pharmacological inhibitors in stroke treatment, specifically focusing on the effects of inhibitors of poly(ADP-ribose) polymerase (PARP) and inducible nitric oxide synthase (iNOS) on stroke outcomes [66]. PARP, a DNA repair enzyme, is known to contribute to ischemic brain damage when overactivated, and its pharmacological inhibition has been shown to mitigate the brain damage caused by MCAO in rodent models [78], while mice deficient in PARP1 exhibit significant protection from ischemic brain injury [79,80,81]. In their study, ischemia was induced by occluding the middle cerebral artery for 20 min in 2–3-month-old C57BL/6 mice, and the resulting injury was quantified 72 h later.

The study found that the PARP inhibitor PJ34 decreased infarct volume and significantly reduced the post-ischemic increase in iNOS mRNA levels by 72% [66]. However, when iNOS inhibitors were administered alongside the PARP inhibitors, the expected neuroprotective effects were negated, revealing a previously unknown detrimental relationship between iNOS and PARP in the context of cerebral ischemia. This interaction was associated with the increased expression of the inflammatory genes, including those coding for iNOS, intercellular adhesion molecule-1 (ICAM-1), and gp91phox, a crucial component of the NOX2 complex [66].

These findings emphasize the complex role that NOX plays in stroke recovery, particularly the intricate balance between iNOS and PARP activity. However, the study’s limitations, including the use of pharmacological inhibitors as a pre-treatment and the short observation period of 72 h, highlight the need for further research to explore the therapeutic potential of targeting NOX activity over extended recovery periods in a clinically relevant timeframe.

## 5. Role of NOX in Stroke with Complications

NOX enzymes play a crucial role not only in ischemic stroke itself but also in stroke recovery complicated by conditions such as hypertension, obesity, and hyperglycemia [67,68,69].

One preclinical study was conducted to address the question of whether to continue or discontinue antihypertensive medication in patients who experience a stroke while already on blood pressure-lowering treatment. In this study, olmesartan, a potent angiotensin II receptor blocker (ARB) with notable cerebral protective properties demonstrated by both experimental and clinical studies in ischemic stroke models, was selected for evaluation [67].

Adult male Sprague–Dawley rats that had undergone permanent MCAO were divided into different groups to assess the impact of olmesartan treatment on stroke-induced injuries, with measurements taken at 3, 7, and 14 days post-stroke. Terminal deoxynucleotidyl transferase dUTP nick end labeling (TUNEL) staining was employed to evaluate cell death, revealing a significant reduction in TUNEL-positive cells in the peri-infarct zone in groups treated with Olmesartan, either before or after stroke, compared to the control animals. This suggests that olmesartan administration can significantly mitigate the cell death associated with ischemic injury [67]. Furthermore, the study also examined the expression levels of NOX4, a marker of oxidative stress with a crucial role in the development of ischemic stroke. Fourteen days after the procedure, all groups showed NOX4 expression in the peri-infarct area, but the MCAO/olmesartan group, in which olmesartan was administered from 7 days prior to and 14 days following infarction, exhibited significantly lower NOX4 levels than the other olmesartan-treated groups. This indicates a strong association between olmesartan treatment and the inhibition of NOX4 expression, consistent with its therapeutic potential in reducing oxidative stress after ischemic stroke [67].

The study demonstrates that olmesartan administration, both before and after stroke onset, significantly contributes to the recovery process by reducing cell death and oxidative stress through NOX4 inhibition. The time-dependent effects observed at 3, 7, and 14 days post-stroke highlight the sustained impact of olmesartan in promoting neuroprotection and aiding in the recovery process. These findings reinforce the therapeutic potential of targeting NOX4 regulation as a key strategy in enhancing stroke recovery.

In the context of obesity-induced diabetes, a significant risk factor for stroke, research has highlighted the interaction between sirtuin (Sirt1) and NOX in stroke recovery. Tulsulkar et al. [68] investigated the effects of a high-fat, high-carbohydrate diet (HFCD) on ischemic damage and recovery processes using a mouse model of permanent MCAO. The study involved C57BL/6 mice, initially aged 8–10 weeks, which were fed an HFCD for 3.5 months, making them 32–34 weeks old at the time of stroke induction. It can be considered a model of cerebral infarction in middle-aged adults. The HFCD diet-induced obesity and hyperglycemia (O/H) mice showed no differences in infarct size and grip strength measured 7 days after stroke compared to normal mice but showed neurological deficits and impaired locomotor activity [68].

The key finding of this study is the identification of the correlation between Sirt1 expression and NOX4 expression in the O/H group after MCAO. Sirt1, a central regulator of metabolic homeostasis and vascular biology, appears to counteract the detrimental effects of a high-fat diet on metabolic health and ischemic stroke outcomes. Its overexpression may facilitate angiogenesis and cerebral remodeling, potentially mitigating the differences in infarct volume between the dietary groups. The study suggests that the increased expression of Sirt1 may be linked to the downregulation of NOX4 in the O/H group post-MCAO, potentially offering a protective mechanism through enhanced angiogenesis and cerebral remodeling [68].

However, it is important to note that despite the observed downregulation of NOX4, the O/H mice exhibited worsened neurological deficits, which suggests that the reduction in NOX4 expression could be involved in the mechanisms that exacerbate these deficits. Given that ROS levels were not measured in this study, it remains uncertain how the decreased NOX4 expression contributes to the overall oxidative stress and neurological outcomes. This study has limitations in that it is difficult to fully understand the role of NOX4 reduction in stroke recovery without ROS monitoring and long-term recovery outcome observations.

A recent study by Won et al. investigated the effects of post-stroke hyperglycemia, which occurs in 30–60% of ischemic stroke patients, on stroke outcomes using a non-diabetic mouse model [69]. Specifically, the study used 3–5-month-old C57BL/6 mice in a photothrombotic cortical stroke model to explore how hyperglycemia, maintained during the interval of 17 to 48 h after ischemia onset, impacts stroke recovery. The effects of post-stroke hyperglycemia were assessed 5 to 25 days after stroke, revealing an increase in the infarct volume, blood–brain barrier disruption, and hemorrhage formation, all of which impaired motor recovery. In addition, the study found that hyperglycemia increased NOX-dependent superoxide formation around the infarct, while *p47phox* mice, which are unable to form NOX2 complexes, did not exhibit the same increase in superoxide formation or motor dysfunction.

These findings indicate that hyperglycemia, occurring from hours to days after ischemia, may amplify oxidative stress in peri-infarct tissues by boosting NOX activity in reactive microglia/macrophages, ultimately leading to poorer functional outcomes [69].

This study suggests that hyperglycemia in the acute phase after stroke can significantly affect subsequent stroke outcomes, highlighting the importance of NOX regulation during this critical period. Therefore, the control of post-stroke hyperglycemia, which can exacerbate stroke outcomes, may be effectively achieved through the modulation of NOX activity. These preclinical findings suggest that targeting NOX in the acute phase of stroke recovery has significant clinical potential, offering a promising avenue for improving outcomes in patients experiencing post-stroke hyperglycemia.

## 6. Discussion

This review highlights the multifaceted roles of NOXs in stroke recovery, emphasizing their complex involvement in various stages of the recovery process, as well as in the context of different therapeutic interventions and comorbid conditions.

### 6.1. Role of NOXs in Angiogenesis and Neurogenesis in Post-Stroke Brain Repair

These studies highlight the pivotal roles of NOX2 and NOX4 in promoting angiogenesis and neurogenesis during stroke recovery [56,57]. NOX2 plays a crucial role in the early stages of vascular remodeling post-stroke, facilitating revascularization and brain repair. Conversely, NOX4’s involvement appears more prominent in the later stages of recovery, influencing neurogenesis and the survival of NPCs. Additionally, NOX1 was identified as a critical regulator of oxidative damage during neurogenesis, where its inhibition was shown to enhance the survival and maturation of neural progenitor cells in the peri-infarct area, leading to improved functional recovery [58]. These findings suggest that the timing and context of NOX activation are critical for their beneficial effects in stroke recovery.

### 6.2. NOX-Mediated Mechanisms in Post-Stroke Functional Recovery

The role of NOXs in post-stroke functional recovery is particularly evident when examining the effects of NOX2, NOX1, and the influence of age [58,59,60,62].

NOX1 inhibition via RNA interference in a stroke model significantly improved motor function recovery, demonstrating the potential of targeting NOX1 to mitigate post-stroke damage. The combined treatment with NADPH and APO showed positive effects on post-stroke recovery by reducing brain inflammation and neuronal damage. However, the timing of APO administration was crucial because, while it was beneficial when administered early, its use in the later stages of recovery disrupted favorable mechanisms, leading to adverse outcomes. Additionally, age-related differences were evident, as older animals exhibited different responses to NOX inhibition compared to younger ones, with aged rats showing worsened outcomes in response to APO treatment.

These studies emphasize the importance of carefully considering the timing, the specific NOX isoforms targeted, and the age of the patient during therapeutic interventions to optimize stroke recovery outcomes.

### 6.3. Implications of NOX Modulation in Stroke Interventions

Several studies have explored the significance of NOX modulation in various stroke intervention strategies, such as physical exercise, electroacupuncture, and pharmacological treatments [64,65,66].

Exercise after reperfusion appeared to improve stroke outcomes by decreasing NOX activity and the expression of NOX subunits, including gp91, and by reducing lactate production. However, these potential benefits are uncertain due to the lack of functional recovery measurements. Electroacupuncture preconditioning, targeting NOX4, demonstrated protective effects on blood–brain barrier (BBB) integrity and reduced oxidative stress, emphasizing the potential of NOX4 inhibition in stroke interventions. However, the short-term evaluation and pretreatment nature of the treatment make the findings uncertain for application to stroke recovery. Pharmacological treatments revealed a complex interplay between NOX activity and stroke outcomes. The combined use of PARP and iNOS inhibitors unexpectedly increased NOX2 expression, exacerbating inflammatory responses and worsening recovery.

These findings highlight the intricate balance required when modulating NOX activity in stroke interventions and the need for precise strategies that consider the specific NOX isoforms, timing, and type of intervention used.

### 6.4. Influence of Comorbid Conditions on NOX Activity in Stroke

The involvement of NOX has been emphasized in the context of stroke complicated by comorbidities such as hypertension, obesity, and hyperglycemia [67,68,69].

The continuous administration of antihypertensive medication may help mitigate functional impairment caused by stroke by inhibiting NOX4 expression. Additionally, in stroke complicated by obesity and hyperglycemia, increased NOX4 expression may be associated with functional deficits, while elevated blood glucose levels after stroke may contribute to increased superoxide production through NOX2 activation.

Understanding the role of NOX in stroke complicated by comorbidities could lead to the development of NOX-targeted therapies. Moreover, monitoring specific NOX isoforms as biomarkers in these pathological conditions may present potential strategies for enhancing stroke recovery.

## 7. Conclusions and Future Directions

This review has demonstrated the diverse roles of NOXs in stroke recovery, highlighting both their potential therapeutic benefits and risks, depending on the specific NOX isoforms involved, the timing of intervention, and the presence of comorbid conditions.

The evidence suggests that while NOX inhibition can mitigate the damaging effects of oxidative stress in the acute phase of stroke, the role of NOXs becomes increasingly complex as the recovery process progresses.

NOX2 appears to exhibit both protective and harmful effects depending on the recovery stage, age, and specific cellular environment. Similarly, NOX2 and NOX4 positively contribute to vascular remodeling after stroke, whereas NOX1 is recognized as a key regulator of oxidative damage in post-stroke neurogenesis, with its modulation being associated with improved functional recovery. Additionally, NOX2 and NOX4 contribute to functional deficits in the presence of comorbid conditions such as hypertension, obesity, and hyperglycemia.

The dynamic nature of ROS levels at various stages of stroke recovery presents both challenges and opportunities for therapeutic intervention. The timing of NOX inhibition is crucial; NOX inhibition is beneficial when ROS levels are excessive, but in other conditions, it may disrupt beneficial redox signaling pathways.

Therefore, further research should focus on better understanding these dynamics to develop targeted strategies that maximize the benefits of NOX inhibition while minimizing potential risks. This highlights the need for comprehensive studies that evaluate the long-term outcomes of stroke recovery.

## Figures and Tables

**Figure 1 antioxidants-13-01065-f001:**
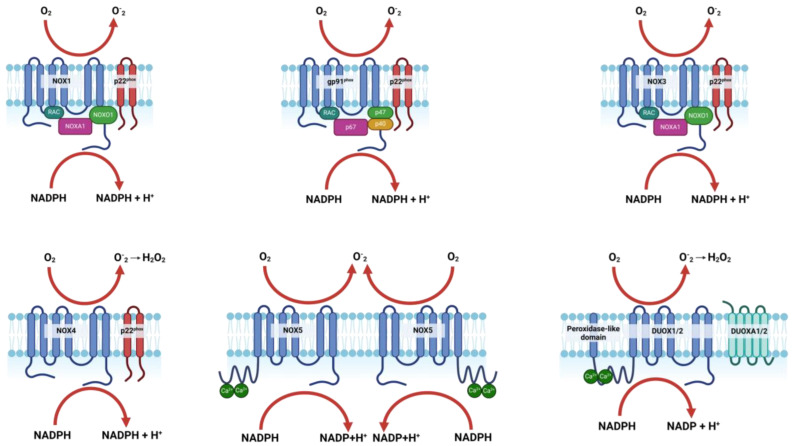
Structure and activation mechanisms of NOX isoforms and subunits. Created with BioRender.com.

**Figure 2 antioxidants-13-01065-f002:**
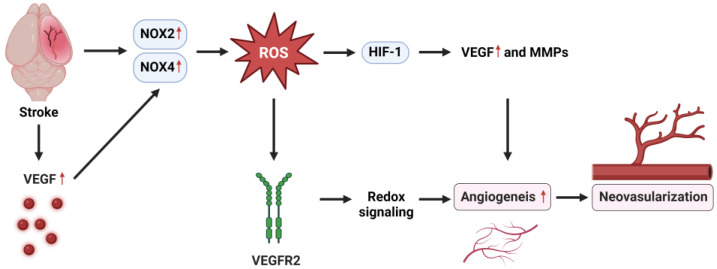
NOX mechanisms in angiogenesis after stroke. VEGFR2: VEGF receptor2; HIF-1: hypoxia inducible factor-1. MMPs: Matrix metalloproteases. The red arrows in the diagram indicate the increase in expression or activation at each step of this pathway.

**Figure 3 antioxidants-13-01065-f003:**
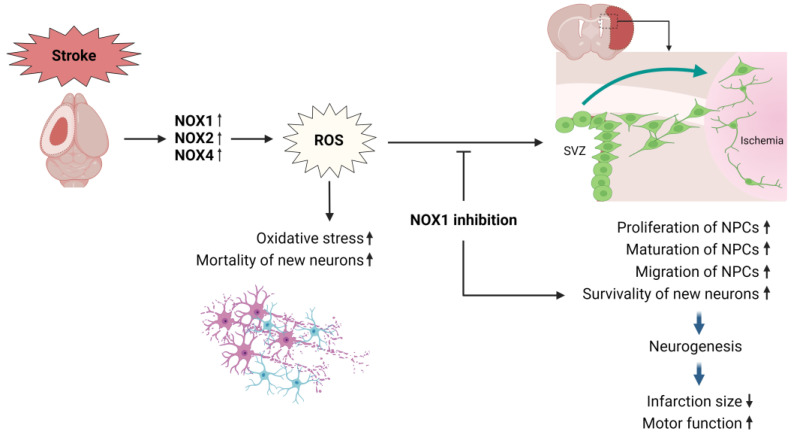
NOX mechanisms in neurogenesis after stroke. SVZ: subventricular zone; NPCs: neural progenitor cell; The upward arrow indicates an increase, while the downward arrow indicates a decrease.

**Table 1 antioxidants-13-01065-t001:** Regulation, activation, and functions of NOX isoforms.

NADPH Oxidases Isoforms	Regulation/Activation			Function
	Activator	Organizer	Regulator	
NOX1	NOXA1	NOXO1	Rac1 GTPase PKC	Superoxide
NOX2	p67phox	p47phox	Rac2 GTPase PKC p40phox	Superoxide
NOX3	NOXA1	NOXO1	Rac1 GTPase	Superoxide
NOX4	-	-	p22phox TGF-β	Hydrogen peroxide
NOX5	Ca^2+^	-	Calmodulin	Superoxide
DUOX1 DUOX2	Ca^2+^ DUOXA1 DUOXA2	-	p22phox	Hydrogen peroxide

PKC: protein kinase C; TGF-β: transforming growth factor beta.

**Table 2 antioxidants-13-01065-t002:** Overview of Animal models, timepoints, interventions, and outcomes in selected NOX-related stroke recovery studies.

Study	Animal Species	Animal Model	Age or Weight	Timepoints	Intervention	Outcomes
Taylor, 2013 [56]	Wistar rats	ET-1 induced MCAO	10–12 w	24, 48, and 72 h, 7, 14, 21 and 28 days	-	Neurological score, Adhesive touch, Adhesive remove, Infarct size
McCann, 2014 [57]	C57BL/6 mice	MCAO	8–12 w	6, 24, and 72 h	*Nox2* KO	Neurological score, Infarct size
Choi, 2015 [58]	Wistar rats	MCAO	8 w	6 h, 1 and, 2 d and 4 w	*Nox1* KD	Parallel bar score, Infarct size
Qin, 2017 [59]	ICR mice	MCAO	25–30 g	8 and 16 h, 4 w	APO	Rotarod, Neurological score, Infarct size
Yingze, 2022 [60]	C57BL/6 mice	MCAO	8–10 w	3, 7, 10, and 14 d	APO	Rotarod, Neurological score, Infarct size
Fabian, 2008 [61]	Wistar rats	CAO	7 d	24 and 72 h	APO	Superoxide
Kelly, 2009 [62]	Sprague-Dawley rats	MCAO	3–4 months; 18–20 months (aged)	24 h	APO	Neurological score, Infarct size
Weston, 2013 [63]	Wistear rats	ET-1 induced MCAO	10–12 w	24, 48, and 72 h	APO	Neurological score, Infarct size
Shen, 2016 [64]	Sprague-Dawley rats	MCAO	280–300 g	30 and 48 h, 4 d	Exercise	Apoptotic cell death
Jung, 2016 [65]	C57BL/6 mice	MCAO	20–25 g	24 h	EA	Neurological score, Infarct size
Park, 2004 [66]	C57BL/6 mice	MCAO	8–12 w	6, 24, and 72 h	iNOS and PARP inhibitors	Infarct size
Gutierrez-Fernandez, 2015 [67]	Sprague-Dawley rats	MCAO	250–320 g	72 h, 7 and 14 d	Olmesartan	Rotarod, Neurological score, Infarct size
Tulsulkar, 2016 [68]	C57BL/6 mice	MCAO	8–10 w	7 d	-	Rotarod, Neurological score, Infarct size
Won, 2024 [69]	C57BL/6 mice	Photothrombosis	12–35 w	−5 to 25 d	gp91ds-TAT, *p47* KO	Superoxide, Skilled reaching task Rotating beam test, Sticky tape test

ET-1: endothelin-1; MCAO: middle cerebral artery occlusion; CAO: carotid artery occlusion; KO: knockout; KD: knockdown; h: hour(s); w: week(s); d: day(s); EA: electroacupuncture; Except for those specifically marked as ‘aged’, all animals were within the young adult age range.

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
