# Peer review of "Role of NADPH Oxidases in Stroke Recovery"

_antioxidants, 2024, doi:10.3390/antiox13091065_

Round 1

Reviewer 1 Report

In this review article, the authors state that they wish to contribute to the field by discussing the role of NADPH oxidases during recovery (not hyperacute, not acute) following stroke.

However, at the moment, the review is a little disjointed and lacks focus.

The authors do not provide a sufficient frame of reference - 16 articles are mentioned at the beginning that they authors would discuss, but many other articles appear to be discussed in parallel.

The authors state that they wish to review recover periods but this is not clear in the subsequent discussion.

In vitro results appear to be discussed alongside in vivo results, without adequate discussion.

Importantly, the authors do not provide an adequate critique of the animal studies and their relevance to the clinic - treatments administered prior to stroke are of much less practical potential in the clinic, for example.

Thus, I have several clarifications and comments for the authors.

Citation1, line 31
Please provide a more appropriate review, e.g., from GBD series, e.g., https://www.thelancet.com/journals/laneur/article/PIIS1474-4422(24)00038-3/fulltext

Citation 6, line 41
Please provide a more up-to-date citation

Citation 10 line 53
Please provide a more up-to-date citation.

Lines 72 - 76
Were the articles limited to human studies?
If human please use Risk of bias tools to assess data e.g., https://www.riskofbias.info/welcome/robvis-visualization-tool
If animal studies please use SYRCLE tools to assess quality of data https://www.syrcle.network/

To properly compare studies, the authors must describe the models or patient criteria - for recovery, what timepoints were used (what was the authors' definition of recovery versus hyperacute or acute effects)?

Line 174
"Several studies..."
Are the authors referring to one of the 16 studies mentioned in 75-76?

A table of the papers, the models used, the timepoints examined, the age of the rodents used and main outcomes would be most helpful, right at the beginning of the review. Then the reader would be able to follow along more easily.

It would be important to discuss how soon after stroke the agents, e.g., APO, should be given - this should be included in a more succinct discussion.

Given that age appears important (line 256-258), the age of all models must be discussed.

Lines 281-283
Here the authors appear to be referring to acute effects, although the aim of the review was to examine remote effects during recovery. Please provide a more focused discussion.

Line 293-295 - here the authors appear to refer to in vitro work - is this one of the 16 citations referred to in lines 75-76?

Section 4.1 - here again, it is unclear why the authors are referring to very early timepoints given their claim to be examining recovery as stated in the introduction lines 62-76

Please clarify why and how acupuncture *prior* to stroke is relevant to the clinic (lines 342-345).

Again, a better focus of the review is required - in section 4.3, the authors now discuss timepoints up to 14 days.

Throughout their review, the authors are asked to sincerely discuss the preclinical work and whether and how it is relevant to the clinic where the the initial 3 months are considered critical.

Section 4.4
The relevance of wound scratching to stroke is unclear. Please justify how this is relevant to the topic.

Timepoints for section 4.5, relevant to the aims of the review (of recovery following stroke), are unclear.
Please discuss.

Table 1
This table appears to mix in vivo and in vitro results, which is confusing. The authors are asked to provide a more precise table, and to place it towards the beginning of the review, with model, age, timepoints, intervention, etc, included.
It is unclear what enhancing (E) suppressing (S) means - and thus, this column is uninformative as it neither supports nor critiques NOX isoforms.
"Factor regulating stroke recovery " is similarly unclear - to what are the authors referring that is relevant to the aims of the review?

The Discussion is far too long and it appears that the authors now refer to other studies.
Thus, again, the focus of this review is unclear.

Should the authors wish to provide an introduction to NOX in stroke (which appears to be covered within section 5), this reviewer would suggest placing it at the beginning of the review.

Reviewer 2 Report

This review is generally well prepared. Moreover, two areas need to be improved:

1) Essential information about a) NOX structure, function and isoforms, and b) mechanisms underlying regulation and activation of NOXs

2) Lack of figures for illustrating NOX role in Brain Repair after stroke and molecular and cellular pathways involved.

To include essential information about a) NOX structure, function and isoforms, and b) mechanisms underlying regulation and activation of NOXs, in the format of text and illustrative figure(s)

To include schematic figures to illustrate NOX role in Brain Repair after stroke and molecular and cellular pathways involved.

Round 2

Reviewer 1 Report

I thank the authors for their sincere attention to all of my comments.

All of my comments have been addressed.

Reviewer 2 Report

The authors have adquately addressed my concerns through their substantial revision.

To do more proofreading for avioding any spelling and grammatical errors prior to your final submission and upload the figures with high resolutions for publication.